# Pushing nano-aerosol measurements towards a new decade – Technical note on Airmodus Particle Size Magnifier 2.0

Juha Sulo[1], Joonas Enroth[2], Aki Pajunoja[2], Joonas Vanhanen[2], Katrianne Lehtipalo[1,3], Tuukka Petäjä[1], Markku Kulmala[,1,4,5]

[1] Institute for Atmospheric and Earth System Research (INAR) / Physics, Faculty of Science, University of Helsinki, Finland
[2] Airmodus Oy, Helsinki, 00560, Finland

[3]Finnish Meteorological Insititute, Helsinki, Finland

[4]Aerosol and Haze Laboratory, Beijing Advanced Innovation Center for Soft Matter Sciences and Engineering, Beijing University of Chemical Technology (BUCT), Beijing, China

[5]Joint International Research Laboratory of Atmospheric and Earth System Sciences, School of Atmospheric Sciences, Nanjing University, Nanjing, China

*Correspondence to*: Juha Sulo (juha.sulo@helsinki.fi)

**Abstract.** Accurate measurement of the size distribution of sub-10 nm aerosol particles is still a challenge. Here we introduce a novel version of the Airmodus particle size magnifier (PSM), which is a condensation particle counter-based instrument with a sizing range of 1 – 12 nm. The extended size range compared to the earlier PSM version enables direct detection of forming clusters and particles as well as studying their growth processes without the challenges related to particle charging. It also gives an overlap between the activation size distribution measurements with the PSM and mobility size distribution measurements with conventional mobility particle sizers. We compared the performance of PSM 2.0 to a mobility particle size spectrometer, the original A10 particle size magnifier and a Neutral Cluster and Air Ion Spectrometer (NAIS) during field measurements. Also, calibration results were compered against the A10 instrument. The results show that PSM 2.0 is able to activate sub-2 nm clusters and the concentration and size distribution between 2-12 nm compare well especially with the NAIS.

## 1 Introduction

Understanding and quantifying the process of new particle formation requires direct observations of the forming particles in the relevant size range, no matter the environment. An estimated 40-80 % of total aerosol number concentration is formed through new particle formation (Gordon et al., 2017; Dunne et al., 2016; Weagle et al., 2018) even in megacities (Kulmala et al., 2021) Therefore understanding this process is important for reducing the uncertainties in aerosol climate effects as well as health effects of nanoparticles. New particle formation (NPF) is a complicated process which is typically characterized by the formation of clusters in the size range of 1 – 2 nm that may under favourable conditions grow further into nucleation mode and finally into climate relevant sizes, potentially acting as cloud condensation nuclei. Current research indicates that the initial

cluster formation and their growth beyond 2 nm are two separate processes (Tröstl et al., 2016; Kulmala et al., 2004) and can include partly different precursor vapours. The formation and growth of the smallest particles from sub-3nm range to 10 nm and beyond are key steps in understanding how many freshly formed particles survive to contribute to the particle loading in the atmosphere. Earlier studies have observed a large number of particles in the sub-3 nm size range e.g. in  fresh traffic emissions (Rönkkö et al., 2017, Lintusaari et al., 2023), boreal forests (Kulmala et al., 2013, Sulo et al., 2021), polluted environments (Kontkanen et al., 2017, Zhou et al., 2020, Kulmala et al., 2021), rainforest (Wimmer et al., 2018), arctic environments (Sipilä et al., 2021), mountaintops (Bianchi et al., 2020)  as well as industry (Ahonen et al., 2017; Sarnela et al., 2015) and indoor air (Farmer et al., 2019).

 The capacity to measure and size aerosol particles routinely down to 1 nm in size has only been developed in the past decade, with instruments such as the Particle Size Magnifier (PSM; Vanhanen et al., 2011) and the Neutral cluster and Air Ion Spectrometer (NAIS; Mirme and Mirme., 2013) bridging the gap between mass spectrometers and particle counters. However, measuring the smallest particles and clusters is a challenging task and each measurement device has its own challenges (Kangasluoma & Kontkanen 2017). These issues are typically related to the way the instruments make particularly the sub-3nm particles detectable, either through charging or condensational growth. Electrical mobility spectrometers suffer from  low charging probabilities of the smallest particles and the fact that charger ions typically are of similar size to the actual sample in the sub-3 nm range (Kangasluoma et al., 2018), while condensational growth methods must achieve a high enough supersaturation of the working fluid to overcome the Kelvin effect, which typically leads to homogeneous nucleation. However, in practise the CPC techniques utilise both heterogeneous nucleation on insoluble particles and nano-Köhler process in the case of soluble particles (Kulmala et al., 2007). Currently, observing particle growth from the 1-2 nm size range up to nucleation and accumulation mode sizes requires a combination of multiple instruments such as in Kulmala et al. (2013) and Stolzenburg et al. 2022. Although some methods for combining particle size distributions from several different instruments exist and have produced excellent results (Stolzenburg et al., 2022), they still require data overlap between the instruments and assumptions about the instrument detection efficiencies and losses. Moreover, this requires a battery of instruments in order to resolve the size distribution of aerosol particles between 1 and 10 nm. Here we introduce an improved version of the Airmodus A10, called PSM 2.0. The new instrument extends the upper size limit from 4 nm up to 12 nm, without losing accuracy or detection efficiency of the smallest particles.

The aim of this article is to introduce the new PSM 2.0 prototype and its conceptual design, showcase its capacity to measure the size distribution of 1 to 12 nm particles and compare its performance with the earlier PSM version (A10) and with mobility particle sizers (NAIS and DMPS) during field measurements. A full characterization of the instrument is subject of future studies.

## 2 Conceptual background

Condensation Particle Counters (CPCs) are commonly used for measuring the total particle number concentration above a certain diameter (cut-off size), which depends mainly on the supersaturation achieved inside the instrument. Aerosol particle sizing with an adjustable supersaturation has been used previously. These include the original particle size magnifier (Seto et al., 1997), the work of Gallar et al. (2006) in building a CPC with variable supersaturation (VSCPS), as well as the original Airmodus PSM (Vanhanen et al. 2011). The required supersaturation for particle activation in a CPC is typically achieved in

one of three ways (McMurry et al., 2000): adiabatic expansion, turbulent mixing or laminar diffusive activation. The design of the PSM 2.0 is a hybrid, which uses turbulent mixing to rapidly mix the sample and saturator flows. However, unlike in a mixing-type CPC, the adiabatic mixing does not activate the particles, but in fact the activation is based on a laminar diffusive supersaturation. To further reduce the mixing type-activation inside the mixing section, the inlet of the PSM is slightly heated (40°C). In this, the working principle of the PSM 2.0 is similar to that of recent version of the VSCPS technology by Attoui et

al. (2023), with their two-flow mixing system.

        The original design of the Airmodus A10 is described in detail by Vanhanen et al. (2011). In that instrument, the size selective particle activation and growth was achieved by varying the saturator flow rate between 0.1 to 1.3 lpm. The inlet flow rate of the instrument was held constant by changing the excess flow rate after the condenser. This varying saturator flow led to a varying total flow rate through the most crucial part of the instrument – the mixing section. With the varying flow rate,

the temperature and the vapor amount also varied causing the instrument to find a dynamic equilibrium, which can be susceptible to instability.

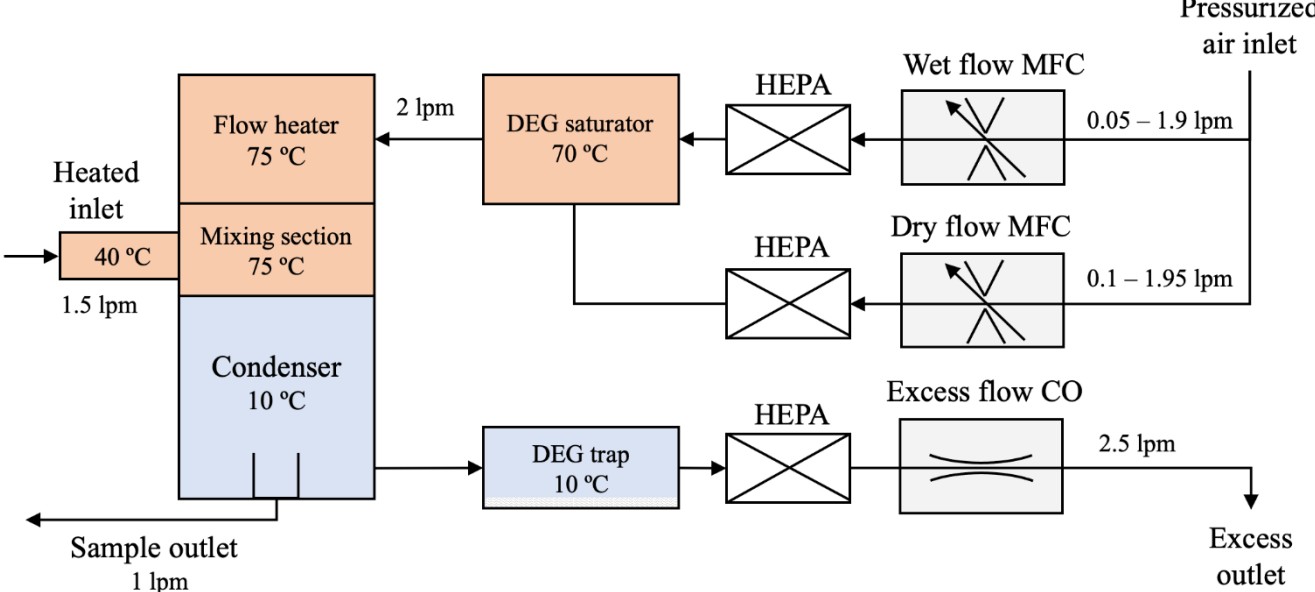

**Figure 1. The flow diagram of the prototype PSM 2.0.**

85        The PSM 2.0, in contrast, has a constant flow rate passing through the mixing section in addition to constant, and temperature controlled, inlet flow. The only parameter which is varied is the ratio of vapor amount arriving to the mixing section from the saturator. The total flow containing both dry and wet flow is referred as the saturator flow. Similarly as in the instrument of Attoui et al. (2023), two mass flow controllers are operated in tandem to produce a constant 2 lpm flow rate. One mass flow controller regulates the amount of dry air reaching the heater prior the mixing section, while the other one

controls the flow passing through the saturator prior entering the same heater element as the dry flow (Figure 1). This means that although the saturator flow rate is defined similarly as in the A10, the flow into the mixing piece is kept constant by the addition of a heated dry flow, thus allowing the supersaturation of the instrument to be varied without changing the flow rate into the mixing piece.

        In the prototype PSM 2.0, the total saturator flow rate has been increased to 2 lpm, of which at most 1.9 lpm can be

wet saturator flow. Also, the inlet flow rate has been decreased to 1.5 lpm. This means that the highest wet saturator flow rate in the instrument has increased almost 60 % compared to the A10, allowing us to reach the same maximum supersaturation with up to 10 °C lower saturator temperature, as shown in Figure 2a. The difference is only partially explained by the difference in flows, and to a large degree by a re-designed saturator. Unlike in the A10, the wet flow is fully saturated even at the highest flow rates, which enables lowering the saturator temperature without compromising on the resulting vapor pressure, and hence

the maximum achieved supersaturation. The lower temperature of the saturator is key for enabling the extended size range of the PSM 2.0 (Figure 2b). It means that the total vapor pressure of DEG after the mixing section is more than five times lower than it is in the A10, allowing for much finer control of the supersaturation and therefore particle activation and growth. The duration of a single scan (240 s) has remained the same as in the A10, but the increased stability allows increasing the time resolution by analysing the up and down scans separately unlike with the A10. A potential further increase in the time resolution

is a subject for future characterization studies. D50 as a function of the wet flow behaves exponentially due to the Kelvin effect, but in the PSM2.0 the behaviour is clearly more linear compared to A10 even though the sizing capability is increased from 1-4 nm to 1-12 nm. D50 is the particle diameter at which 50% of the particles are activated on a given flow rate. As shown in Vanhanen et al (2011) and elsewhere, the particle activation in A10 is not fully consistent with ideal mixing type activation. The design of the PSM 2.0 was intentionally moved further away from the mixing type activation, and towards

diffusive activation. The temperature of the heater is held intentionally at a higher temperature than the saturator. This both prevents the condensation of DEG into the heater, but also ensures that even in the case of adiabatic mixing the temperature after the mixer is sufficiently high to keep the saturation ratio below unity. The aerosol carrier flow is still turbulently mixed with the saturator flow as in the A10, but here the turbulent mixing is only a method to achieve a uniform vapor pressure across the aerosol sample flow efficiently and rapidly. This also reduces the diffusional losses of sample compared to more laminar

designs.

The size selective particle activation is dependent on the homogeneity of the supersaturation field across the aerosol sample and reflected e.g. in the steepness of the cut-off curve. As shown by previous work, there are two commonly used approaches in continuous flow instrument to achieve a sharp cut-off curve. One is to use adiabatic expansion, but this results in only semi-continuous sampling. The second, and more common solution, is to use a core fed aerosol flow together with a vapor sheath

which will result in the aerosol experiencing a more homogenous supersaturation field across its streamlines. Here, we use the opposite approach of this method. The design of the PSM condenser is such, that it is effectively performing core sampling of the saturated flow mixed with the aerosol. While this design "wastes" both aerosol sample and DEG vapor, it is easy to control and results in an equally sharp cut-off curve. Hence, we can achieve sharper cut-off response than in a single flow instrument (Figure 2c, which is required for the extended size range. A regular CPC is used for subsequent particle detection. In this study,

the CPC used was an Airmodus A20.

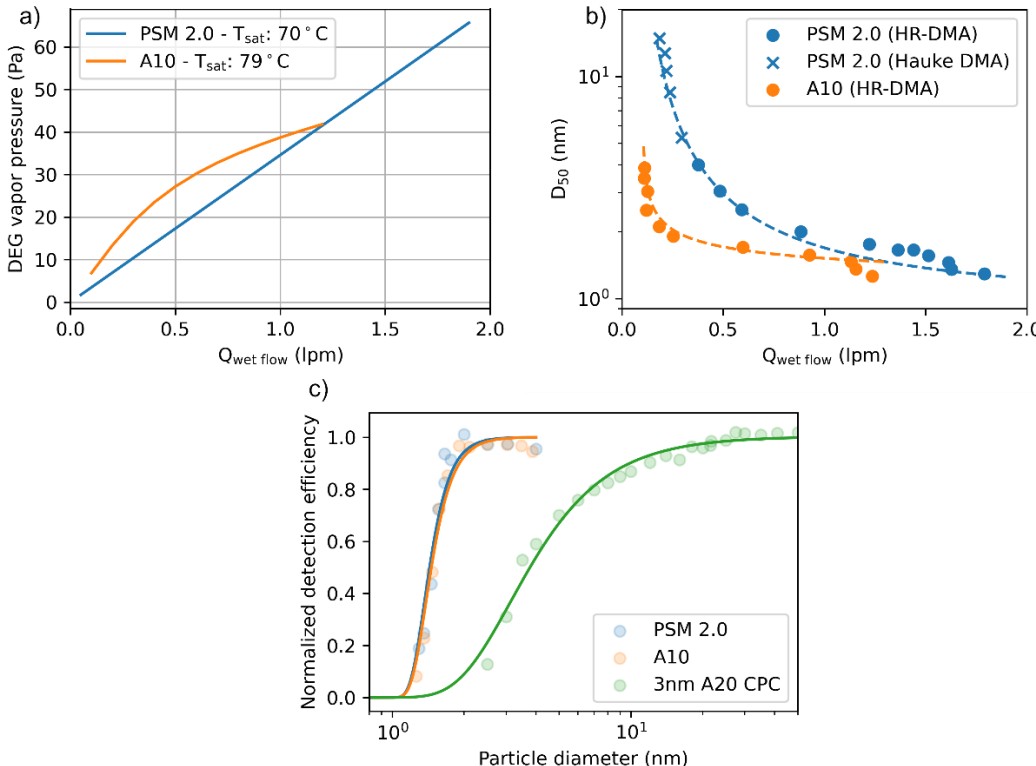

**Figure 2. a) Modelled DEG vapor pressure at the mixing section of the A10 and PSM 2.0. b) Measured calibration curve of the PSM 2.0 prototype and an original A10. The increased linearity in the vapor pressure, as a function of the saturator flow enables the PSM**

**2.0 to have a larger activation diameter range than the A10. D$_{50}$ is the particle diameter at which 50% of the particles are activated on a given flow rate. c) Cut-off curves for the highest saturator flow for the A10 and prototype PSM 2.0, and a 3 nm cutoff version of the full flow A20 CPC, showing the considerable steeper cutoff-curves achieved with the PSM technology.**

## 3 Results and discussion

The cut-off curve (detection efficiency as a function of size) for PSM2.0 compared to A10 PSM and A20 CPC is shown in Fig 2c. The results show that the lowest cut-off size of PSM2.0 is comparable to PSMA10, and for both of them the cut-off size is much lower than for the A20 CPC with a nominal 3 nm cut-off size. The calibration setup used for the A10 and PSM 2.0 calibration consisted of two distinct configurations. For the sub-5 nm particles, we used the NiCr particles, size selected with the half-mini (Fernández de la Mora & Kozlowski, 2013) that was calibrated with tetra-heptyl ammonium bromide mobility

standards. As our setup did not permit larger particles than 4.5 nm to be selected with this setup, we used a Hauke type differential mobility analyzer (e.g. Winklmayr et al. 1991) to produce monodisperse silver particles up to 12 nm.

To test the performance of prototype PSM2.0 in field conditions and to compare it to other instruments measuring the particle number concentration and size distribution, we performed side-by-side measurements at the SMEARIII station in

Helsinki during May 2023. SMEAR III is an urban background station located at the Kumpula Campus area ca. 4 km from the Helsinki city center (Järvi et al., 2009). The inversion method used to resolve aerosol particle number size distributions was the step-wise method (Lehtipalo et al., 2014). The inversion methods used for the A10 are not yet fully optimized for the PSM 2.0 and full development of the inversion methods, including kernel and EM methods, is a subject of further study. Initial inversion results were able to resolve 12 size bins in the 1 – 12 nm size range.


First, we investigated how well the PSM2.0 is able to detect the smallest atmospheric particles and ions. In Figure 3 we show the total concentration of sub-2 nm particles measured by the PSM 2.0 as well as its predecessor A10 and compare those to the concentration of sub-2 nm air ions (sum of positive + negative cluster ions) measured by the Neutral Cluster and Air Ion Spectrometer (NAIS; Mirme and Mirme, 2013). We observe that during nighttime PSM 2.0 measured concentrations

similar to the ion concentrations measured with the NAIS, showing that it is able to activate most of the atmospheric cluster ions. During daylight (the sunrise in Helsinki in May is around 4am and sunset 8:30 pm local winter time), the concentration measured by PSM2.0 was higher than the cluster ion concentration, probably due to neutral clustering of low volatility vapors that form due to photochemical reactions (Jokinen et al. 2017), although contribution from traffic cannot be excluded. We were also able to capture a new particle formation event during midday, characterized by a sudden clear increase of the

sub-2 nm particles, showcasing the instrument's sensitivity at both large and small concentration ranges. The concentration measured by A10 PSM was clearly lower than the PSM2.0, probably due to limitations in the activation efficiency of the smallest particles. It has been seen also earlier that the A10 PSM does not detect all of the atmospheric sub-2 nm ions, although it is sensitive to sub-3nm particles formed especially during new particle formation (Sulo et al. 2021), as seen also in Figure 3. Uncertainties in Figure 3 and subsequent figures were calculated by determining the systematic uncertainties for

each instrument such as the amplitude of random fluctuations in CPCs and uncertainty in electrometer currents and including random uncertainty in charging and sizing by using a Monte Carlo simulation similarly as in Kangasluoma et al., (2020).

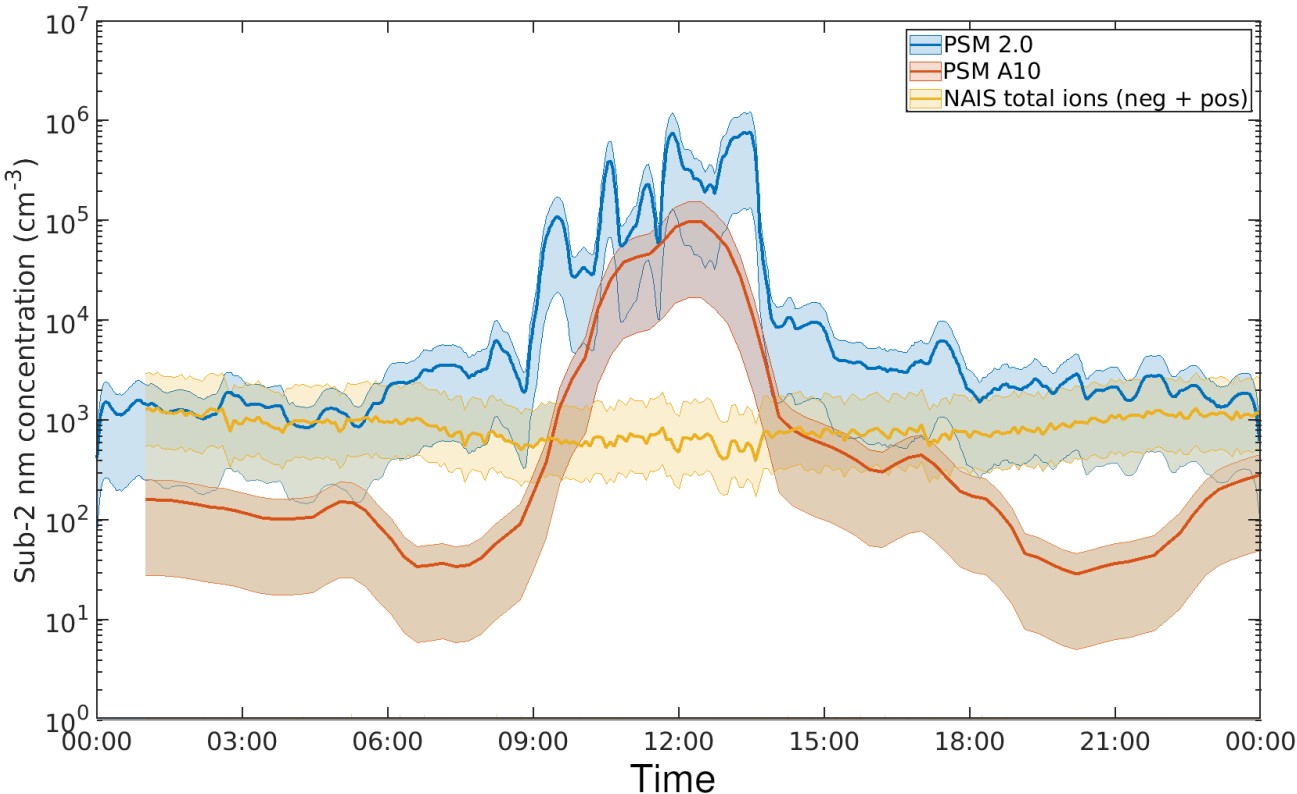

**Figure 3: Cluster concentrations in sub 2 nm size range observed with PSM2.0 and A10 PSM in blue and yellow, respectively, and total sub-2 nm ion concentrations measured by NAIS in purple during May 9th 2023. The concentration uncertainties were calculated in the same way as in Kangasluoma et al., (2020).**

The 2 – 7 nm size range is often considered the key size range for new particle formation (Leino et al., 2016; Tammet et al., 2014) and in particular the key size range for determining whether freshly formed particles grow into climate-relevant sizes. Our results (Figures 4a, S1) show that the prototype PSM2.0 was measuring very similar total concentration in the 2.5 – 7 nm size range as the NAIS particle mode, without the need for charging the sample. In the–7 - 12 nm size range (Figure 4b, S2) the PSM2.0 was agreeing rather well with the differential mobility particle sizer (DMPS). The NAIS was consistently seeing somewhat higher concentrations than the DMPS and PSM2.0. This feature has been seen in many previous instrument comparisons (Gagne et al. 2011; Kangasluoma et al. 2020), and more sophisticated DMPS systems tuned to measure ultrafine particles (e.g. those used in Kangasluoma et al., 2020) could provide even better agreement with the PSM 2.0. The differences in the DMPS and PSM2.0 -measured concentrations are likely due to reduced counting statistics in the DMPS

(Stolzenburg et al., 2023). During the new particle formation event with higher concentrations in this size range (ca 12-14 pm) all of the instrument agreed well.

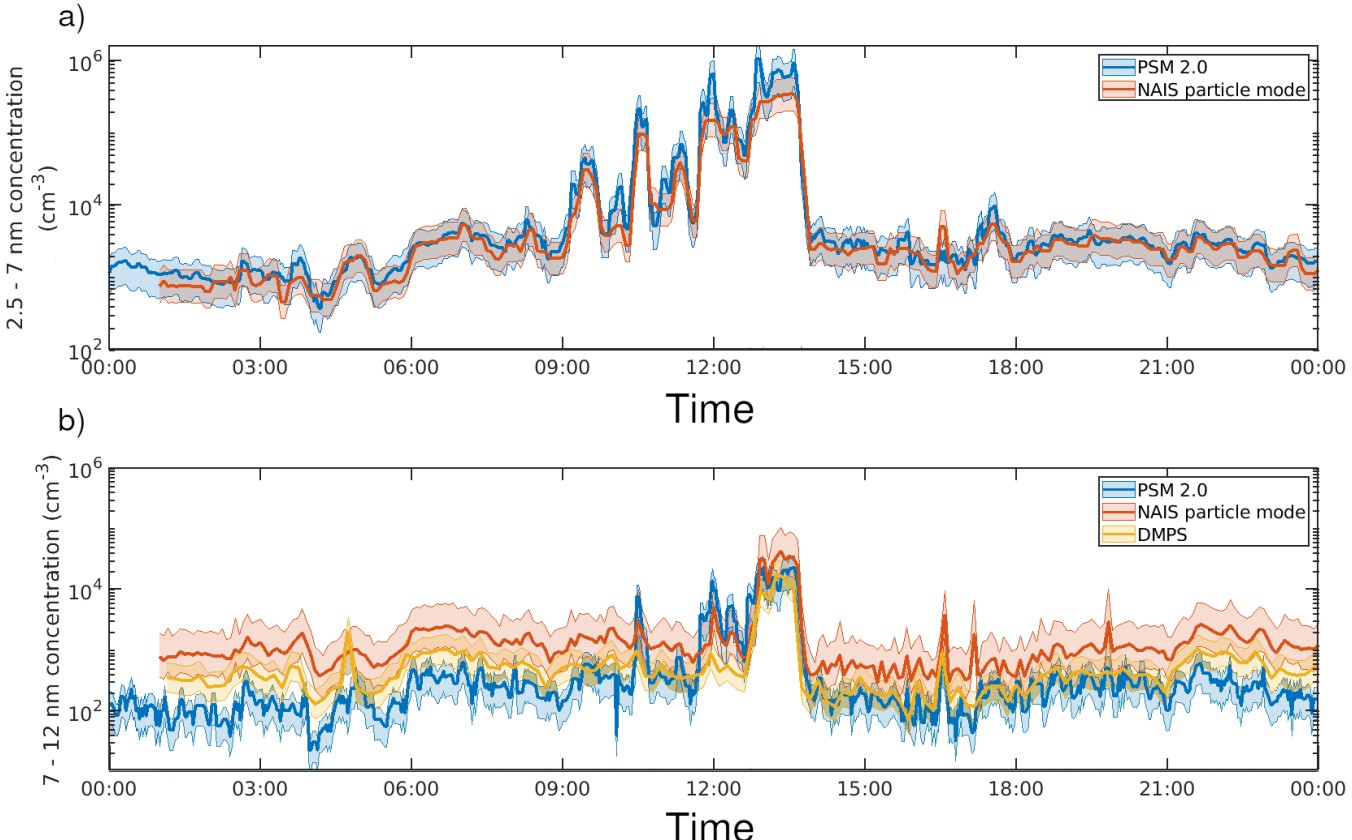

**Figure 4. a) The concentration of 2.5 – 7 nm aerosols measured by the PSM 2.0 and NAIS in particle mode. b) The concentration of**
**7 – 12 nm concentrations measured with the PSM 2.0, NAIS in particle mode and DMPS. The concentration uncertainties were calculated in the same way as in Kangasluoma et al., (2020).**

Finally, we investigated the whole size distribution of 1 – 12 nm particles during and outside the new particle formation event time (Figure 5). Our measurements show a rather good agreement between the PSM2.0 and the size distributions

measured by the other instruments, especially considering that the PSM determines particle size based on totally different principle (particle activation) than the mobility particle sizers. Most notable differences in the size distribution are seen in the 2 – 4 nm size range. The 2 – 4 nm size distribution is known to be very difficult to measure accurately, as most instruments

suffer from significant losses and/or low charging probabilities and solving the size distribution in this range includes large uncertainties (Kangasluoma et al., 2016, Kangasluoma et al., 2020). Figure 5 also shows that although the total particle

number concentration in the the 2.5 - 7 nm size range is very similar between NAIS and PSM 2.0, there exists differences between the number concentrations of different size particles within that size range as measured by NAIS or PSM 2.0. The drop in the NAIS particle mode concentration close to 2-3 nm could also be related to the instrument post-filter designed to remove ions formed in the corona charger (Manninen et al. 2016) and a similar cross-over (i.e. NAIS showing more than other instrument above 3 nm, but less between 2-3 nm) has been seen also in earlier instrument comparisons (Kangasluoma et al. 2020). The DMPS is seeing clearly lower concentrations than the other instruments below ca. 5 nm during the event and below ca. 8 nm during night. All in all, it is likely that the improved stability and sensitivity of the PSM 2.0 allows us to more accurately detect the particle size distribution below 4 nm than previously.

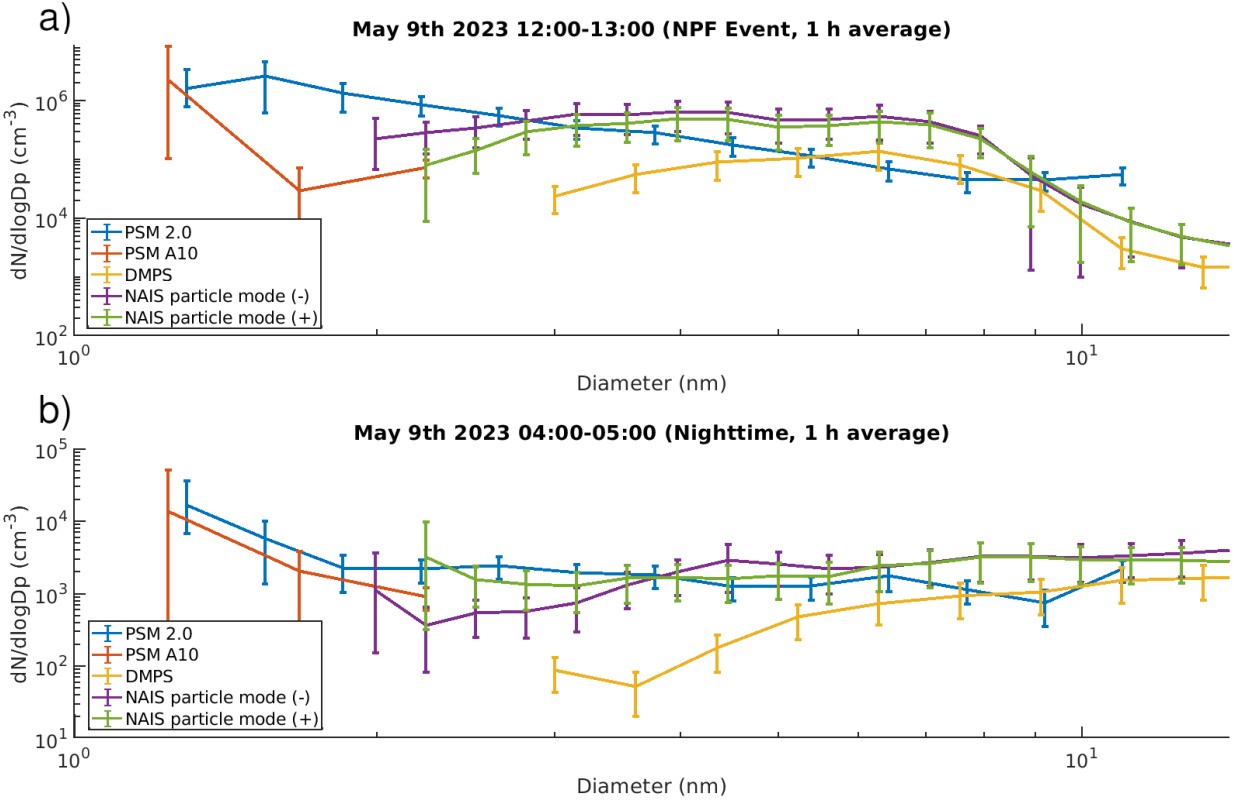

Figure 5: **The size distribution measured with PSM2.0, A10 PSM, DMPS, and NAIS particle mode in negative and positive polarizations during NPF event (a) and during nighttime (b). The concentration uncertainties were calculated in the same way as in Kangasluoma et al., (2020).**

## 4 Conclusions

Here we presented a new prototype version of the Airmodus Particle Size Magnifier, namely PSM2.0. The new instrument is capable of detecting both sub 2 nm clusters, intermediate particles (2-7 nm) and also the lower end of the nucleation mode

(7-12 nm). It is significantly more effective in activating clusters in sub 2 nm size range than the previous PSM versions, e.g. it is able to observe in practise all small ions during night-time and also enhanced concentrations of sub 2 nm particles during NPF events, and therefore it improves the closure between predicted and observed cluster concentrations (Kulmala et

al. 2022). Furthermore, the number concentrations of 2-7 nm particles measured by PSM2.0 and NAIS agreed very well, although the detailed size distributions show some differences between different instrument types. These results are consistent with earlier instrument comparisons (Kangasluoma et al. 2020; Stolzenburg et al. 2022) showing the need of instrumentation optimized for the sub-10 nm size range

Additionally, PSM2.0 should be significantly more stable than PSM A10 due to the redesign of the saturator flow system, flow mixing and internal liquid handling, which allows for finer control of the supersaturation and reduces uncertainty from turbulent mixing and non-uniform supersaturation. This improves its applicability in field conditions. Proper characterization of the instrument stability is a subject of future work. This work should include studies about the sizing accuracy, seed material dependency especially in the larger sizes and the time resolution limits of the PSM2.0. It is currently the only

instrument capable of measuring the whole size range relevant for particle formation and early growth without needing to charge particles. Therefore, we can state that PSM2.0 could become revolutionary in the investigation of the dynamics of sub 10 nm clusters and particles.

**Data availability**

The data used is available in https://zenodo.org/doi/10.5281/zenodo.10663799 (Sulo, 2024). The data are licensed under a
Creative Commons 4.0 Attribution (CC BY) license.

**Author contributions**

JE and JV designed the instrument, JE, and JV and AP participated in the laboratory calibration. JE did the modeling, JS and AP participated in the field measurements. JS, JE and JV performed the data analysis, while JS, JE, JV, KL and MK wrote the paper. TP and MK facilitated access to research infrastructure. All authors contributed comments to the article.

**Competing interests**

MK is member of the editorial board of the Aerosol Research journal. KL, MK, JE, JV and AP are shareholders of Airmodus Oy. JE, JV and AP work for Airmodus Oy.

**Acknowledgments**

We acknowledge the following projects: ACCC Flagship funded by the Academy of Finland grant number 337549 (UH) and
337552 (FMI), Academy professorship funded by the Academy of Finland (grant no. 302958), Academy of Finland projects

no. 1325656, 311932, 334792, 316114, 325647, 325681, 347782, "Quantifying carbon sink, CarbonSink+ and their interaction with air quality" INAR project funded by Jane and Aatos Erkko Foundation, "Gigacity" project funded by Wihuri foundation, European Research Council (ERC) project ATM-GTP Contract No. 742206. University of Helsinki support via ACTRIS-HY is acknowledged. Support of the technical and scientific staff in Kumpula are acknowledged

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
