# Peer review of "Pushing nano-aerosol measurements towards a new decade – Technical note on Airmodus Particle Size Magnifier 2.0"

_Aerosol Research, 2023_

## Author Comment (AC1)

A substantial improvement of a commercial particle sizing instrument in the nm range - the A10 PSM - is described which consists in the extension of the size range to 1-12 nm and the increase in detection efficiency. The PSM 2.0 provides important overlap to other sizing instruments. The optimization of the flows in the instrument prevents shortcomings of the precursor instrument such as instabilities due to variability of the saturator flow. The mixing of sample flow and saturated flow was optimized by keeping the total flow to the mixing point constant but with different ratios of saturated and dry air. Finally, a size calibration and a field comparison with other instruments, among them mobility based instruments, is presented and the effect of charging in the bespoke size range is discussed.

1. Does the paper address relevant scientific questions within the scope of AR? YES
2. Does the paper present novel concepts, ideas, tools, or data? YES
3. Are substantial conclusions reached? YES
4. Are the scientific methods and assumptions valid and clearly outlined? YES, see comments.
5. Are the results sufficient to support the interpretations and conclusions? PARTIALLY, see comments.
6. Is the description of experiments and calculations sufficiently complete and precise to allow their reproduction by fellow scientists (traceability of results)? PARTIALLY, see comments.
7. Do the authors give proper credit to related work and clearly indicate their own new/original contribution? YES
8. Does the title clearly reflect the contents of the paper? PARTIALLY, see comments.
9. Does the abstract provide a concise and complete summary? YES
10. Is the overall presentation well structured and clear? YES
11. Is the language fluent and precise? YES
12. Are mathematical formulae, symbols, abbreviations, and units correctly defined and used? PARTIALLY, see comments.
13. Should any parts of the paper (text, formulae, figures, tables) be clarified, reduced, combined, or eliminated? YES, see comments.
14. Are the number and quality of references appropriate? YES
15. Is the amount and quality of supplementary material appropriate? No supplementary material was provided.

Answer: We appreciate the reviewer's detailed comments and have amended the manuscript accordingly. The consistent terminology and the added discussion on methods and future work will help clarify the message of the manuscript.

**detailed comments:**

- Title: For a technical note paper speaking of "a new decade" is a bit overdone and early in my point of view. The below discussion shows that some aspects such as time resolution are not yet properly characterized. Matter of taste of course.
  o Answer: We have amended the title slightly: "Pushing nano-aerosol measurements towards a new decade – Technical note on Airmodus Particle Size Magnifier 2.0"

- From line 84 on and also in Fig. 1 the PSM 2.0 design and work principle is outlined. I miss several details here which should make it easier to comprehend the pros of the new design, moreover, the nomenclature is not consistent. Is the "mixing section" in line 86 the same as the "T-block" in Fig. 1? There is some inconsistency in the description of the flows in the instrument: Lines 86-87: "…two mass flow controllers are operated in tandem to produce a constant **2 lpm** flow rate…". I understand this in the way that 2 lpm are constantly fed into the T-block. Lines 90-91: "In the prototype PSM 2.0, the total saturator flow rate has been increased to **2 lpm**, of which at most 1.9 lpm can be wet saturator flow." So, what are actually the saturator flow, dry flow, and outlet flow values? These should be described. What is meant by "wet saturator flow"? Is it the flow downstream of the saturator? Having mentioned that, the following statement in Line 90f is not easy to understand: "This means that the ratio of saturator flow to total flow in the instrument has increased almost 60 % compared to the A10….". The term "Total flow" is not defined. If it is saturator flow plus inlet flow, in my understanding for the A10 this is a varying quantity while in PMS 2.0 it is described as constant. The authors mentioned possible temperature instabilities in the A10. However, the temperature control in the PSM 2.0 is not sufficiently described. The temperatures or temperature ranges of inlet, flow heater and condenser should be described. Is there a need or an option to cool the inlet flow under hot ambient conditions?
  - Answer: The terminology has been made consistent by using the term "mixing section" in both text and figures. We have added a description of the total flow as "The total flow containing both dry and wet flow is referred as the saturator flow." and listed the flow rates of different parts of the instrument in Figure 1. Additionally, we have included the temperatures of the inlet, flow heater and condenser in Figure 1. Because heating of the inlet is done in order to minimize the mixing type-activation, there should be no need to cool the inlet flow under ambient conditions.
- I would like to ask for comments about possible effects of sample heating at the inlet of the PSM 2.0.
  - Answer: We have added an explanation for the heating of the PSM inlet: " To further reduce the mixing type-activation inside the mixing section, the inlet of the PSM is slightly heated (40$^{\circ}$C)"
- There is another inconsistency in terminology: Line 112 uses the term "PSM growth tube", this is obviously the "Condenser" in Fig. 1. I recommend to thoroughly check the manuscript for consistent language and terminology.
  - Answer: The terminology has been amended as per the reviewer's comment.
- Line 85 states that in the PSM 2.0 "the only parameter which is varied is the ratio of vapor amount arriving to the mixing section from the saturator". In Fig. 2a and 2b, A10 and PSM 2.0 are compared, and the abscissa of the diagrams is QReaders may question why the diagrams are not plotted as functions of the ratio. $Q_{saturator}$ is probably defined differently for the PSM 2.0 and the A10 and the reader cannot comprehend how the "ratio of vapor" depends on $Q_{saturator}$.
  - Answer: The saturator flow rate is defined the exact same way, but in PSM 2.0 the addition of the dry flow keeps the flow rate into the mixing piece constant, increasing stability. We have added a text clarifying this: "This means that although the saturator flow rate is defined similarly as in the A10, the flow into the mixing piece is kept constant by the addition of a heated dry flow, thus

allowing the supersaturation of the instrument to be varied without changing the flow rate into the mixing piece."

- It is probably beyond the scope of this paper, but I miss some information on the time resolution of the new PSM 2.0. Is there a gain in time resolution of the PSM 2.0 by faster reaching stability after changing saturator and dry flows?

  o Answer:We have added a mention of the time resolution: "The duration of a single scan (240 s), during which the wet flow is increased from 0.05 to 1.9 lpm and back down to 0.05 lpm, has remained the same as in the A10, but the increased stability allows analysing the up and down scans separately and thereby increasing the time resolution. A potential further increase in the time resolution by reducing the scan time is a subject for future characterization studies. D50 as a function of the wet flow behaves exponentially due to the Kelvin effect, but in the PSM2.0 the behaviour is clearly more linear and less steep compared to A10 even though the sizing capability is increased from 1-4 nm to 1-12 nm (Figure 2b)."

- As to field measurements and Figs. 3, 4 and 5, statements on concentration uncertainties: "The concentration uncertainties were calculated in the same way as in Kangasluoma et al., (2020)." This puzzled me a bit because Kangasluoma (2020) very comprehensively discussed factors of uncertainty but did not present any calculations; instead, a Monte Carlo simulation of uncertainties was presented. It should be explained in more detail if and how this simulation was used for the uncertainty calculations of the data presented. I refrain from further commenting on the field measurements because of lack of expertise in this field of atmospheric research.

  o Answer: The uncertainties were calculated by determining the systematic uncertainties for each instrument, including random fluctuations in CPC counts or FCE measured current, as well as transfer function width and resolution. On top of this the random uncertainty in charging and sizing are estimated by using a Monte Carlo simulation. We have added a text to clarify this: "Uncertainties in Figure 3 and subsequent figures were calculated by determining the systematic uncertainties for each instrument such as the amplitude of random fluctuations in CPCs and uncertainty in electrometer currents and including random uncertainty in charging and sizing by using a Monte Carlo simulation similarly as in Kangasluoma et al., (2020)."

The particle size magnifier (PSM) was a major advance for aerosol formation studies, as it unveiled the sub-3 nm size range at an unprecedented level (enabling neutral cluster detection at high sensitivities). However, the PSM as presented back in 2011 had major shortcomings: 1) Its operation was from time to time cumbersome and required very experienced researchers for best data output, 2) its counting efficiency was highly seed-material dependent introducing major uncertainties and 3) its size range was limited preventing efficient comparison with other instruments and 4) its time resolution was rather low. Sulo et al. present the PSM 2.0, which promises to solve 1) and 3). The PSM 2.0 is certainly an important update and has the potential to become an essential instrument in sub-10 nm particle studies. The paper is well-written, concise, and certainly presents novel insights suitable for publication in AR. However, in this technical note I am missing a discussion of 2) and 4). Moreover, I think that the improvements on 1) are also insufficiently discussed. Therefore, I can only recommend publication of the manuscript after the following minor points have been revised.

Answer: We thank the reviewer for their insightful comments. We have added clarifications to the manuscript as described below under each detailed comment from the reviewer.

- 1 misses some important details. It would be helpful if the flow arrows show the typical flow rate ranges (or fixed values) as already written in the text. Also, all heated/cooled segments should be labelled with the default temperature settings. For the condenser I would illustrate the separation between sample and excess flow via core-sampling somehow also in the Figure. The DEG-trap is not mentioned in the text (what is it, why is it there and how does this improve the PSM2.0?) and while the reader familiar with the PSM might not wonder, I think it is necessary to mention that a CPC is used for subsequent particle detection, and it should be mentioned what type was used in this study.

  o Answer: We labeled the heated and cooled segments in Figure 1 and added the typical flow rates in each segment to make the figure more informative as the reviewer correctly points out. We have also clarified the separation between sample and excess flow. The DEG trap was part of the original design of the A10 and its function is to condense and collect the DEG in the excess flow so that it can be drained from the instrument. We have added additional information into the text regarding the DEG trap and mentioning the use of a regular CPC after the PSM and included information on the one used in this study: "After the condenser the sample is taken to a regular CPC for subsequent particle detection. In this study, the CPC used was an Airmodus A20. The DEG trap in the excess flow is used to condense excess DEG and drain it via the drain valve."

- As the authors claim "Additionally, PSM2.0 is significantly more stable than PSM A10 due to (…)" in their conclusion I ask them to provide additional information on how they proof this claim. This could be e.g., long-term time-series of both instruments, stability of background during several scans, overall drift of regular background measurements (as performed with an Airmodus Diluter for example, see Lampimäki et al., 2023, J. Aerosol Sci.) or variability with external conditions (such as humidity). I don't request the authors to perform more experiments, but just to support the claim of improved stability to some extent, maybe with the

data they already have at hand. Otherwise, the statements on improved stability need to be removed.

o Answer: The reviewer is correct, that the increased stability is yet to be properly demonstrated. We have modified the text to: " Additionally, PSM2.0 should be significantly more stable than PSM A10 due to the redesign of the saturator flow system, flow mixing and internal liquid handling, which allows for finer control of the supersaturation and reduces uncertainty from turbulent mixing and non-uniform supersaturation. This improves its applicability in field conditions. Future work to characterize the instruments should include studies about its stability in field conditions, sizing accuracy, seed material dependency and optimizing the time and size resolution."

• In the instrument description I am missing information on scan-time and procedure and the inversion procedure applied. How many size-bins can be obtained from the PSM 2.0 within the 1-12 nm range, or what resolution is expected (The steepness of Fig. 2b lets me think that it might be quite tricky to resolve above 5 nm into too many bins)?

o Answer: We have specified the inversion method used and added additional commentary on it: "We used the step-wise method by Lehtipalo et al. (2014) to resolve the aerosol particle number size distributions. However, the inversion methods developed for A10 are not yet fully optimized for PSM 2.0 and the development of a new inversion method and optimizing the amount of size bins is a subject of further studies. The initial inversion method was able to resolve 12 size bins in the 1 – 12 nm size range."

• Can the authors already comment on the see-material dependence of the PSM2.0 at larger sizes? If not, they should at least mention that this is an important subject for future studies. They should also shortly explain why the plateau of the detection efficiency now reaches 1 (and never reached this value in the A10). This also greatly reduces the total concentration uncertainties obtained from PSM2.0.

o Answer: Since the PSM2.0 uses the same working fluid (diethylene glycol) as A10, we expect that there might be some seed-material dependency. However, this is not yet properly explored, especially at larger sizes. The seed-material dependence is a topic of further studies and we have added a mention of that into the conclusions: "Future work to characterize the instruments should include studies about its stability in field conditions, sizing accuracy, seed material dependency and optimizing the time and size resolution." As the reviewer correctly points out, there are differences in the plateau values between the instruments, but the plateau of the detection efficiency contains some uncertainties still for the PSM 2.0 calibration, and we have therefore adjusted the plots to have normalized detection efficiencies in order to avoid confusing the reader. Further characterization of the detection efficiency plateau value is a subject of future work.

• I think Fig. 4 and Fig.5 with their more than four orders of magnitudes in concentration on the y-axis do not fully enable a judgement on the concentration agreement. Can you add a plot which shows the ratio between the instruments over time in the SI? Moreover, it should be mentioned that Fig.4a most likely shows such a good agreement between NAIS and PSM2.0 for the wrong reasons. NAIS underestimates below 3 nm but overestimates above 4 nm, which altogether

might be the reason for the good agreement. In fact, the comparison shown in Fig. 5 is very close to the results from Kangasluoma et al. 2020 who used more sophisticated sub-10 nm DMPS systems. In that sense, the PSM2.0 seems to possibly provide pretty good agreement with such sub-10 nm DMPS systems, and I would love to see a comparison to one of those instruments in the future. With respect to that it could also be mentioned that the reduced agreement with the "standard"-DMPS below 5 nm is most likely due to reduced counting statistics as recently shown by Stolzenburg et al., 2023, Atmos. Meas. Tech.

- Answer: We have added plots which show the ratios between the instruments over time to the SI. Additionally, we have commented on the reasons for the good agreement in the 2.5 – 7 nm particle number concentration by stating "Figure 5 also shows that although the total particle number concentration in the 2.5 - 7 nm size range is very similar between NAIS and PSM 2.0, there exists differences in the size distribution within that size range. The drop in the NAIS particle mode concentration close to 2-3 nm could also be related to the instrument post-filter designed to remove ions formed in the corona charger (Manninen et al. 2016) and a similar cross-over (i.e. NAIS showing more than other instrument above 3 nm, but less between 2-3 nm) has been seen also in earlier instrument comparisons (Kangasluoma et al. 2020). " and added a mention of the reasons for DMPS concentrations differences by amending the text to: " This feature has been seen in many previous instrument comparisons (Gagne et al. 2011; Kangasluoma et al. 2020), and DMPS systems optimized to measure ultrafine particles (e.g. those used in Kangasluoma et al., 2020) could provide even better agreement with the PSM 2.0. The differences in the DMPS and PSM2.0 -measured concentrations are likely due to reduced counting statistics in the DMPS (Stolzenburg et al., 2023)"

- Last, I think the title of the manuscript might be a bit too provocative. If it really pushed aerosol measurements to a new decade needs to be decided in a decade.

- Answer: Based on this and the other reviewer's similar comment, we have amended the title slightly: "Pushing nano-aerosol measurements towards a new decade – Technical note on Airmodus Particle Size Magnifier 2.0"

Technical:

- Please label x-axis in Fig.3 and 4 with "Time".
- Answer: These have been added.
- 2: Please explain $D_{50}$ in the caption.
- Answer: We have added a sentence to the caption explaining $D_{50}$: $D_{50}$ is the particle diameter at which 50% of the particles are activated on a given flow rate.
- Check that you name condenser and growth tube consistently.
- Answer: This has been checked and amended to "condenser".
- In line 70 it is already stated that the activation occurs in the condenser and only later this statement (line 99) is supported by a reference.
- Answer: we are unsure as to what the referee is referring to. In line 70 we refer to CPC technology in general and provide a reference to McMurry et al., 2000. The description of the

hybrid design in line 74 refers to Attoui et al., (2023). Later descriptions of the activation refer to Vanhanen et al., 2011, such as  in line 99 in the original manuscript (now line 109).

- Line 127: Delete "of" between "setup" and "used"

o   Answer: This has been amended.

---

## Author Response (AR2)

Technical corrections:

line 97: Please change "...ratein..." to "...rate in...".

line 108: Please explain "D50" also in the main text when first mentioning it.

line 185: Please add "Stolzenburg et al. 2023" citation in the list of references.

lines 332-335: Please correct the "Lehtipalo et al. 2014" entry.

Supplement Figure S1b and Figure S2a: Please add at least a second y-axis tick label to clarify the meaning of the y-axis values in Figures S1b and S2a.

Answer: We appreciate the editor's technical corrections and have made the necessary modifications to the manuscript to reflect the high quality of the journal. We have also made small adjustments to the legends in Figures 3 and 4 to make them more colorblind friendly.

line 97: Please change "...ratein..." to "...rate in...".

Answer: We have corrected this grammatical error.

line 108: Please explain "D50" also in the main text when first mentioning it.

Answer: The editor is correct in that we have forgotten to explain D50 in the text. We have added an explanation to the text by modifying the text to: D50 is the particle diameter at which 50% of the particles are activated on a given flow rate.

line 185: Please add "Stolzenburg et al. 2023" citation in the list of references.

Answer: This has been added to the list of references.

lines 332-335: Please correct the "Lehtipalo et al. 2014" entry.

Answer: This formatting error has been fixed.

Supplement Figure S1b and Figure S2a: Please add at least a second y-axis tick label to clarify the meaning of the y-axis values in Figures S1b and S2a.

Answer: We have added further tick labels on the y-axis where there was only a single tick in figures S1b and S2a.